# Tspan8-Tumor Extracellular Vesicle-Induced Endothelial Cell and Fibroblast Remodeling Relies on the Target Cell-Selective Response

**DOI:** 10.3390/cells9020319

**Published:** 2020-01-29

**Authors:** Wei Mu, Jan Provaznik, Thilo Hackert, Margot Zöller

**Affiliations:** 1School of Public Health, Shanghai Jiao Tong University School of Medicine, Shanghai 200025, China; 2Department of General, Visceral and Transplantation Surgery, Pancreas Section, University of Heidelberg, 69120 Heidelberg, Germany; 3EMBL Genomics Core Facility, 69117 Heidelberg, Germany

**Keywords:** tumor exosomes, tetraspanin 8, endothelial cells, fibroblasts, message transfer, mRNA, ncRNA, non-transformed target remodeling

## Abstract

Tumor cell-derived extracellular vesicles (TEX) expressing tetraspanin Tspan8-alpha4/beta1 support angiogenesis. Tspan8-alpha6/beta4 facilitates lung premetastatic niche establishment. TEX-promoted target reprogramming is still being disputed, we explored rat endothelial cell (EC) and lung fibroblast (Fb) mRNA and miRNA profile changes after coculture with TEX. TEX were derived from non-metastatic BSp73AS (AS) or metastatic BSp73ASML (ASML) rat tumor lines transfected with Tspan8 (AS-Tspan8) or Tspan8-shRNA (ASML-Tspan8kd). mRNA was analyzed by deep sequencing and miRNA by array analysis of EC and Fb before and after coculture with TEX. EC and Fb responded more vigorously to AS-Tspan8- than AS-TEX. Though EC and Fb responses differed, both cell lines predominantly responded to membrane receptor activation with upregulation and activation of signaling molecules and transcription factors. Minor TEX-initiated changes in the miRNA profile relied, at least partly, on long noncoding RNA (lncRNA) that also affected chromosome organization and mRNA processing. These analyses uncovered three important points. TEX activate target cell autonomous programs. Responses are initiated by TEX targeting units and are target cell-specific. The strong TEX-promoted lncRNA impact reflects lncRNA shuttling and location-dependent distinct activities. These informations urge for an in depth exploration on the mode of TEX-initiated target cell-specific remodeling including, as a major factor, lncRNA.

## 1. Introduction

Extracellular microvesicles (EV) are of utmost importance in intercellular crosstalk in health and disease [1,2,3]. However, the mode of action is still disputed, particularly whether the transferred EV content acts autonomously or activates host cell-inherent programs [4,5]. We approached this question analyzing the impact of tumor cell-derived small extracellular vesicles (TEX) on endothelial cells (EC) and fibroblasts (Fb). TEX contributing to angiogenesis [6,7] and premetastatic niche preparation, frequently involving Fb of metastasis-prone organs [8,9,10], is the topic of particular interest.

Exosomes are a subpopulation of small 40–100nm EV, which distribute throughout the body and are recovered in all body fluids [11,12]. Exosomes are build-up by a transmembrane protein-containing lipid bilayer and proteins, coding and noncoding (nc)RNA, and DNA in the vesicle lumen. Exosome lipids show higher lipid order and stability against detergents and are enriched in sphingomyelin, cholesterol, GM3/GRM6^1^, and PS^1^, high level PS expression allowing differentiation from microvesicles [13]. Improved mass spectrometry (MS) facilitated exosome protein characterization. Constitutive exosome proteins are structural vesicle components or are involved in vesicle biogenesis and traffic. Most abundant are tetraspanins, 7- to 124-fold enriched in exosomes compared to parental cells. Adhesion molecules, proteases, major histocompatibility complex (MHC) molecules, HSP^1^, TSG101^1^, Alix^1^, annexins, cytoskeleton proteins, metabolic enzymes, cytosolic signal transduction molecules, and ribosomal proteins, some recruited via their association with proteins engaged in biogenesis, also are copious [14,15].

Exosome biogenesis starts with early endosomes (EE) formation, deriving from the trans-Golgi network or internalized membrane microdomains [16]. EE are guided towards multivesicular bodies (MVB) [17]. During inward budding of so-called intraluminal vesicles (ILV) into MVB, vesicles receive their cargo. LPAR1^1^ together with Alix/PDCD6IP and HSP70 promote inward budding. SGPP1^1^ and DAG^1^ are engaged in cargo sorting [18]. Loading of the exosome plasma with proteins, coding and ncRNA, and DNA are nonrandom processes [19,20]. After ILV inward budding, MVB are guided for degradation towards the proteasome or the plasma membrane, the latter involving microtubules and Rab^1^ proteins [17,21]. SNARE^1^ and SYT^1^ are engaged in fusion with the plasma membrane [17,22]. The released vesicles are called exosomes. Exosome components are function competent [23] and content delivery suffices for target cell reprogramming [24,25].

Exosomes bind and are taken-up by selected target cells. Exosome binding frequently involves (tetraspanin-associated) integrins, and among others ICAM, fibronectin (FN), laminins (LN), proteoglycan-binding lectins and PS binding TIM1, -3, -4^1^, GAS6^1^, MFGE8^1^, STAB1^1^, ADGRB1^1^, RAGE/AGER^1^, galectins, selectins and sialic acid binding lectins [26]. Exosome binding is greatly facilitated by protein clusters in both exosomes and target cells [27]. Exosomes are taken-up by fusion with the cell membrane or endocytosis. Exosome ingestion can proceed via phagocytosis, macropinocytosis, clathrin-dependent, and/or tetraspanin-enriched microdomains (TEM), lipid rafts and caveolae [28]. The uptaken exosomes can modulate target cells directly or by stimulating signaling, transcription and silencing processes [29,30], which we elaborated for TEX targeting EC or Fb.

Tetraspanins, most abundantly enriched in exosomes, are highly conserved 4-transmembrane proteins with a small and a large extracellular loop [31]. The latter accounts for dimerization and association with non-tetraspanin partner molecules, tetraspanins associating with a large variety of transmembrane proteins including integrins, cell adhesion molecules (CAM), receptor tyrosine kinases (RTK), proteases, but also cytoskeleton and cytosolic signal transduction molecules [32,33]. Accordingly, TEM function as signaling platforms [34,35,36,37]. TEM are prone for internalization, maintaining the tetraspanin webs after fission and scission [38,39]. Tetraspanins contribute to EE traffic towards MVB partly via an endosomal sorting complex required for transport (ESCRT)-independent pathway [40,41]. Juxtamembrane cysteine palmitoylation provides a link to cholesterol and gangliosides and protects from lysosomal degradation [41,42], which adds to tetraspanin enrichment in exosomes. Finally, the exosome tetraspanin web strengthens binding avidity by clustering tetraspanin-associated molecules [27,43].

These general tetraspanin features also account for Tspan8, a metastasis-associated tetraspanin, upregulated in several malignancies and associated with metastatic spread and poor prognosis [44]. Significant inhibition of colon cancer growth by anti-Tspan8 in vivo without affecting proliferation and apoptosis resistance in vitro strengthens the suggestion of its engagement in intercellular communication [45]. In line with the importance of TEX Tspan8 and associated molecules, Tspan8-α6β4 promotes migration and lung metastasis, but prohibits angiogenesis [46,47,48]. On the other hand, TEX Tspan8-α4/α5 induces disseminated intravascular coagulation (DIC) [49,50]. In vitro, TEX of a Tspan8 overexpressing non-metastatic rat pancreatic adenocarcinoma (PaCa) line (AS-Tspan8) promotes EC progenitor maturation, expansion, and activation. Changes in RNA and protein profiles of EC/EC progenitors being observed after 1 h–5 d coculture [50], suggested EC reprogramming predominantly relying on target cell activation.

We here explored the impact of AS- and AS-Tspan8-TEX on EC and Fb focusing on changes in target cell mRNA and ncRNA profiles. EC and Fb respond differently to AS- and AS-Tspan8-TEX. These distinct responses do not correlate with the TEX content, are target cell-dependent and predominantly engage signaling cascade activation.

## 2. Material and Methods

Cell lines: BSp73AS (AS) is a non-metastatic pancreatic adenocarcinoma line of the BDX rat strain [51]. AS-Tspan8 (formerly D6.1A) were obtained by transfecting AS with the Tspan8 cDNA [49]. BSp73ASML (ASML), the metastatic subline of the same tumor, highly expressing Tspan8, and an ASML-Tspan8 knockdown (kd) line are described [52]. A BDX Fb line was generated by NiSO4 treatment of minced lung tissue cultures [46]. An EC line was derived from cultured aortic cells of a Wistar rat (Cell-lining, Berlin, Germany). These lines were maintained in RPMI1640/10%FCS/glutamine/antibiotics, containing 0.2 mg/mL G418 for the AS-Tspan8 and 0.5 mg/mL G418 for the ASML-Tspan8kd lines. 293T cells (Cell bank, China) were cultured in high-glucose Dulbecco’s modified Eagle’s medium/10%FBS/antibiotics (Gibco, CA, USA). The human PaCa line A818.4 and an A818.4-Tspan8kd are described [53,54]. All cultures were kept in a humidified atmosphere at 37 °C, 5%CO_2_ in air. Confluent cultures were split 1:3. Cells were regularly checked for mycoplasma contamination by a fluorescence detection Kit (ThermoFisher, Germany).

TEX preparation: Subconfluent cultures (AS/AS-Tspan8: ~8 × 10^6^, ASML/ASML-Tspan8kd: ~1 × 10^7^) were cultured for 48h in 8mL RPMI1640 in the absence of FCS. Culture supernatants were cleared (2 × 10min, 500 *g*, 1 × 20min, 2000 *g*, 1 × 30min, 10000 *g*, 4 °C), filtered (0.22 µm) and centrifuged (Beckman Coulter ultracentrifuge, Type 45 Ti rotor, 50 mL, 120 min, 100000 *g*, 4 °C). The resuspended pellet was centrifuged (PBS, 120 min, 100000 *g*, 4 °C). The pellet was resuspended in 0.8 mL and mixed with 0.8 mL 80% sucrose and layered at the bottom of 4 mL ultracentrifugation tubes and overlaid with 1.6 mL 30% and 0.8 mL 5% sucrose and centrifuged (Beckman Coulter ultracentrifuge, SW41Ti rotor, 4 mL tubes, 16 h, 100000 *g*, 4 °C), collecting 12 fraction of 320 µL, TEX being enriched in fractions 1–4 (light density fractions, d: 1.15–1.56 g/mL) [55]. Protein concentrations were determined by Bradford. TEX were stored at −80 °C or used immediately for mRNA/miRNA sequencing.

mRNA deep sequencing (DS) and miRNA microarray: TEX were pretreated with RNAse to eliminate unspecifically attached RNA. Both cell and TEX mRNA and miRNA were extracted using mRNA and miRNA extraction kits according to the supplier’s suggestion (Qiagen, Hildesheim, Germany). For microarrays, the platform was Agilent Technologies Scanner G2505C US45103013, the Agilent array DesignID is 070154 (https://www.agilent.com/en/rat-microrna-microarray). For DS, the Illumina HiSeq 2000 sequencer was used. Sequencing was done in single-end 50 bp fashion. Sequencing was performed at EMBL Genomics Core Facility, Heidelberg, Germany. miRNA data are deposited at GEO (http://www.ncbi.nlm.nih.gov/geo/query/ acc=GSE120185). mRNA data are deposited at ENA database (accession No: PRJEB25446). Mean values of normalized data (Agilent Feature Extraction Software, STAR aligner version 2.5.2a) were compared. Differential recovery was defined by ≥1.5- or ≥2-fold changes in mean signal strength of normalized data.

mRNA and miRNA analysis: Ingenuity pathway analysis (IPA) (www.Ingenuity.com) was used for correlating miRNA with mRNA expression after predicted target selection (http://www.microrna.org, http://www.targetscan.org). Search Tool for the Retrieval of Interacting Genes/Proteins (STRING) (http://string-db.org) databases were used for network analysis [56]. The STRING database (version 9.05) is based on known and predicted protein-protein interactions. Only genes that expression were altered by TEX treatment were chosen to create the graphs, which indicate physical protein interactions and functional associations. Network edges were defined either by confidence (line thickness) or by molecular actions (predicted mode of action) and are mostly based on a medium confidence score. To reduce noise, the networks of our data were simplified by deletion of (i) proteins that were not key proteins and (ii) linker proteins that were linked to other linker proteins or cascades. Cascade classification/significance of overrepresented edges was performed by PANTHER (http://pantherdb.org), KEGG (KyotoEncyclopedia of Genes and Genomes) (http://www.kegg.jp), Reactome (https://reactome.org), and UniProt Keywords. The annotation information of protein-protein interaction (PPI) enrichment *p*-values are included.

Real-time PCR (qRT-PCR): Real-time polymerase chain reaction was performed using a standard TaqMan PCR kit protocol on an Applied Biosystems 7900HT Sequence Detection System (Applied Biosystems). The 10 µL PCR included 0.67 µL of reverse transcriptase product, 1× TaqMan Universal PCR Master Mix (Applied Biosystems), 0.2 µM TaqMan probe, 1.5 µM forward primer, and 0.7 µM reverse primer. The reactions were incubated in a 384-well plate at 95 °C for 10 min, followed by 40 cycles of 95 °C for 15sec and 60 °C for 1min. All reactions were run in triplicate [57]. Primers are listed in Appendix A. Statistical analysis was done by the Δ*C*_T_ method (^Δ^*C*_T_ = *C*_T_ test gene − *C*_T_ endogenous control; ^ΔΔ^*C*_T_ = ^Δ^*C*_T_ sample − Δ*C*_T_ calibrator). For RQ (relative quantification/fold change) wild type (wt) cells or TEX were used as reference.

Western blot (WB): Lysates (cell lysate 30 µg, TEX lysate: 10 µg) (30 min, 4 °C, HEPES buffer, 1% Lubrol or 1% TritonX-100, 1mM PMSF, 1mM NaVO_4_, 10mM NaF, protease inhibitor mix) were centrifuged (13000 *g*, 10 min, 4°C), mixed with antibody (Appendix A) (1 h, 4 °C) and incubated with ProteinG-Sepharose (1 h). Washed lysates, dissolved in Laemmli buffer, were resolved on 10%–12% SDS-PAGE. After protein transfer, blocking, blotting with antibodies, blots were developed with enhanced chemiluminescence (ECL) WB-detection-reagent.

Cytokine array analysis: Fb were cocultured with AS-Tspan8- and EC with AS- and AS-Tspan8-TEX for 48 h. Equal volumes of Fb or EC lysates were incubated with the precoated Proteome Profiler array membrane (Rat Signaling or Rat Cytokine Antibody Array kit; R&D Systems, Minneapolis, MN, USA) and processed according to the manufacturer’s instructions. Signaling molecule and cytokine expression was evaluated using Image J software (National Institutes of Health, Bethesda, MD, USA). Data are presented as relative signal intensity compared to untreated Fb or EC (relative signal intensity = 1).

Matrigel invasion: Cells, in the upper part of a Boyden chamber (RPMI/0.1%BSA), were separated from the lower part (RPMI/20%FCS) by 5 μm pore size polycarbonate-membranes. After 16 h at 37 °C, 5%CO_2_ in a humidified atmosphere, cells were removed from the upper membrane side and the lower membrane side was stained (crystal violet) for light-microscopy documentation.

EC growth and tube formation: The tube formation assay, used to assess the effect of miRNA inhibitors on EC, utilizes the autonomous EC response of self-organization into microvessels (MILLIPORE, Billerica, MA, USA). The 48-well plates were coated with cold matrigel (120 μL/well) and incubated for solidification at 37 °C in 5% CO_2_ in air for 2 h. EC were transfected with miRNA inhibitors and 5 × 10^4^ cells were seeded on the pre-coated matrigel and incubated with conditioned media at 37 °C for 8 h. The microRNA inhibitors for miR-181a, miR-146b, and the negative control inhibitor (Appendix A) were purchased from Exiqon (Vedbaek, Denmark).

Luciferase assay: The dual-luciferase Reporter Assay kit, an optimized system including all reagents for the sequential assay of firefly and Renilla luciferase activity was used (Qiagen, Shanghai, China). The wild type 3′-UTR (untranslated region) of the BTG2^1^ gene with putative binding sites for miR-146b was cloned into the pMIR-luc-UTR reporter plasmid (Addgene, Watertown, MA, USA). miRNA-146b and control mimics were purchased from RiboBio Co, Guangzhou, China. The pRL-TK vector (Addgene, Watertown, MA, USA) provides constitutive expression of Renilla luciferase and served as internal control for normalization of the transfection efficiency. 293T cells were transfected with the dual plasmid vectors and miRNA (100 nM/well) for 24 h in 12-well plates. Cells were washed and incubated in DMEM medium for 48 h. Luciferase activity was measured with Synergy 2 Multi-detection Microplate Reader (BioTek Instruments, Inc., San Diego, CA, USA) using Renilla luciferase for normalization [57]. Each experiment was repeated twice with quintuplicate samples.

Statistics: IBM SPSS software (IBM, New York, NY, USA) was used for statistical evaluation. *p*-values are derived from two-tailed Student’s *t* test, analysis of variance, *p*-values < 0.05 were considered significant. However, for microarray and DS analysis only ≥1.5-fold or ≥2.0-fold differences were taken into account.

## 3. Results

Tumor cell-derived EV (TEX) contribute to angiogenesis and premetastatic niche formation, where Fb and EC distinctly respond to AS- versus AS-Tspan8-TEX [46,50,52]. These distinct Tspan8-/Tspan8 complex-TEX-promoted responses of non-transformed cells appeared well suited unraveling the mode, whereby AS- and AS-Tspan8-TEX affect EC and Fb, particularly whether the response corresponds to the TEX content or relies on TEX-promoted target cell autonomous program activation and whether Tspan8-TEX exert selective activities. Our strategy is outlinesd in the flow diagram (Figure 1).

### 3.1. The mRNA and miRNA Profile of Endothelial Cells, Fibroblasts, and AS-Tspan8-TEX

A prerequisite for analyzing the impact of TEX on Fb and EC was the awareness of the two targets’ native state composition as well as of TEX, supposed to reprogram target cells. Thus, we started comparing the RNA and miRNA profile of EC, lung Fb, and TEX. An overview of the results is presented in the supplement.

The mRNA profile of EC, Fb, and TEX was evaluated by DS (ENA database, accession No: PRJEB25446). Roughly 25% from >20000 mRNA displayed a signal strength of >1000 in EC, Fb, and AS-Tspan8-TEX, the 50 most abundant mRNA being shown (Appendix A). Panther tool analysis revealed no significant differences between the three mRNA preparations in molecular functions, indicating a dominance of binding and catalytic active mRNA (Appendix A). Less than 5% of mRNA differed ≥2-fold in EC versus Fb, the 50 mRNA with the strongest difference being listed (Appendix A). Molecular function analysis pointed towards a slight preponderance of EC in binding and catalytic activity and, less pronounced, of Fb in transcriptional regulator activation (Appendix A). Differences in mRNA levels were more pronounced between TEX and cells, with >25% AS-Tspan8-TEX mRNA exceeding EC and Fb mRNA by >2-fold, mRNA displaying a 10-fold difference are shown (Appendix A). No significant differences were seen in the distribution according to molecular functions (Appendix A).

Besides mRNA, TEX miRNA was frequently reported being of major importance in target modulation. miRNA was evaluated in EC, as well as AS- and AS-Tspan8-, ASML- and ASML-Tspan8kd-TEX and cells using Agilent miRNA arrays (deposited at GEO, accession No GSE120185). 

We started with the comparison of AS-Tspan8-TEX and cell miRNA. From the top 50 miRNA, 35 were recovered in cells and TEX (Appendix A). Searching for significant differences between AS-Tspan8-TEX versus cells (signal strength ≥500, ≥2-fold difference) unraveled a higher number of more abundant miRNA in cells (47) than TEX (6), including several let-family miRNA, described to be frequently more abundant in TEX than cells [58] (Appendix A). Comparing AS- versus AS-Tspan8-TEX (signal strength ≥500, ≥2-fold difference) uncovered 15 distinct miRNA in the top ranking 50 miRNA (Appendix A) and higher recovery of 18 miRNA in AS-, but of 30 miRNA in AS-Tspan8-TEX (Appendix A). The more frequent higher recovery in AS-Tspan8- than AS-TEX might indicate an engagement of Tspan8 in TEX recruitment.

The hypothesis was controlled comparing miRNA recovery in Tspan8-expressing ASML-TEX versus ASML-Tspan8kd-TEX. Lower expression was more frequent in ASML-Tspan8kd- than ASML-TEX (Appendix A). Notably, at a lower signal strength (≥200), 27 miRNA were higher in both ASML- and AS-Tspan8-TEX than ASML-Tspan8kd-TEX and AS-TEX. The reverse, a lower signal strength in both ASML-Tspan8kd-TEX and AS-TEX accounted only for 10 miRNA (Appendix A). Though confirming a slight impact of Tspan8 on miRNA recovery in TEX, we recently elaborated that we are dealing with an indirect effect due to Tspan8 associating with proteins that are directly engaged in miRNA recruitment into ILV [54].

Finally, aiming to evaluate the impact of TEX on nontransformed cells, it became important to know about differences in miRNA levels. This is shown for AS-Tspan8-TEX versus EC miRNA. With 14 of the 50 most abundant miRNA differing between AS-Tspan8-TEX and EC, we concluded that differences are sufficient for TEX impact evaluation (Appendix A).

Taken together, AS- and AS-Tspan8-TEX differ from EC and Fb in mRNA and miRNA composition. In addition, AS- and AS-Tspan8-TEX display differences in mRNA and miRNA. The distinct mRNA and miRNA recovery in TEX-treated target cells provided a solid base for evaluating the impact of AS- and AS-Tspan8-TEX on EC and Fb, which was facilitated by the comparably small number of distinct mRNA and miRNA recoveries.

### 3.2. Tspan8 and TEX Targeting and Uptake

TEX targeting and uptake are of central importance approaching the mode of target modulation by TEX. Having already abundantly reported on the contribution of Tspan8, only previous results are briefly summarized. TEX-Tspan8-selective binding relies on the Tspan8 web, Tspan8-α4/α5β1 complexes preferentially bind EC [50], and Tspan8-α6β4 complexes particularly contact Fb [46,52]. The Tspan8 contribution relies on clustering ligands [48]. The concurrent multiple target cell ligand interactions strengthen binding avidity and facilitate TEX uptake [27,38] (Appendix A).

### 3.3. TEX–Target Cell Interaction and the Overall Impact on Target Cells

To screen for the impact of TEX on targets cells, EC and Fb were cocultured for 2d–3d with 30 µg/mL AS- or AS-Tspan8-TEX. mRNA and miRNA were isolated and subjected to DS (mRNA) or miRNA arrays.

Recovery of 122 Fb mRNA was >2-fold increased after coculture with AS-TEX and of 587 after coculture with AS-Tspan8-TEX, only 49 upregulated mRNA being shared by coculture with AS- and AS-Tspan8-TEX, which implies 538 mRNA upregulation being AS-Tspan8-specific. Similar findings accounted for reduced mRNA expression after coculture; 133 mRNA were reduced by AS-, 557 by AS-Tspan8-TEX and 69 by both TEX (Appendix A, Figure 2A). In EC, 217 mRNA recoveries were increased after coculture with AS-Tspan8-TEX, but only 15 after coculture with AS-TEX. Reduced mRNA recovery also was mostly restricted to AS-Tspan8-TEX coculture (AS-Tspan8-TEX: 110, AS-TEX: 33), the majority of AS-TEX affected mRNA (22/33) being also reduced in AS-Tspan8-TEX-treated EC (Appendix A, Figure 2B). The 50 most strongly affected mRNA in Fb and EC by AS-Tspan8-TEX are shown (Appendix A). Despite the striking difference in the response to AS- versus AS-Tspan8-TEX, according to Panther analysis, molecular function-associated Fb and EC genes were affected at a comparable frequency with possibly a slight increase of catalytic and transcription regulating genes in AS-Tspan8-TEX-treated Fb and EC and of structural molecules in Fb (Appendix A). 

Having clarified a stronger impact of AS-Tspan8- than AS-TEX on Fb and EC, there remained the question on a direct transfer of the TEX content, which was analyzed correlating higher AS-Tspan8-TEX mRNA with upregulated mRNA in AS-Tspan8-TEX-treated Fb and EC. Including mRNA with a signal strength of >1000, from 1199 mRNA higher in AS-Tspan8-TEX than Fb, 107 (8.9%) became upregulated by coculture. From 953 mRNA higher in AS-Tspan8-TEX than EC, 23 (2.4%) were upregulated after coculture with AS-Tspan8-TEX. However, 231 (Fb) and 119 (EC) mRNA became upregulated by TEX treatment, despite being lower in TEX than the target cells (Figure 3A). With regard to the contribution of miRNA, we searched for mRNA that expression was lower in AS-Tspan8-TEX than target cells and that expression became decreased by coculturing Fb or EC with AS-Tspan8-TEX. Similar to upregulated mRNA, low mRNA recovery rarely correlated with recovery in AS-Tspan8-TEX (Figure 3B).

Though rare, reduced mRNA recovery after coculture with TEX might reflect an abundant miRNA transfer. The hypothesis was controlled by comparing up- and downregulated miRNA in AS- and AS-Tspan8-TEX-treated EC with the miRNA level in TEX. Taking a signal strength of 500 in at least one of the populations and a 2-fold change as limits, AS-TEX miRNA levels corresponded with the impact on EC in 3/7 (higher recovery after coculture) and 7/10 (lower recovery after coculture) and with AS-Tspan8-TEX in 2/5 (higher recovery after coculture) and 11/12 (lower recovery after coculture) (Figure 3C–F). Although reduced miRNA recovery in TEX-treated EC frequently correlated with the recovery in TEX, this finding does not argue for a transfer of TEX miRNA shaping the TEX-treated target and higher miRNA recovery in TEX-treated EC rarely correlated with TEX miRNA recovery.

Altered mRNA recovery in TEX-treated Fb and EC poorly correlates with recovery in TEX and the molecular functions of TEX-treated targets partly differ from that in TEX. Higher miRNA recovery in TEX-treated EC is rarely observed in AS- and AS-Tspan8-TEX-treated EC and poorly correlates with the recovery in TEX. Although a reduction in miRNA was more frequently seen and apparently correlated with the recovery in TEX, the findings argue against transferred TEX miRNA shaping the TEX-treated target.

In brief, coculture with AS- and far stronger with AS-Tspan8-TEX modulates Fb and EC. There is no evidence that target cell modulation reflects TEX mRNA or miRNA profiles, pointing towards target cell inherent program activation. This assumption is strengthened by the notion that the responses of Fb and EC are only partly overlapping.

Having elaborated a stronger impact of AS-Tspan8- than AS-TEX on nontransformed cells that is supported by TEX-Tspan8 binding and uptake, but not related to TEX content transfer, we proceeded with a detailed, separate analysis of AS- /AS-Tspan8-TEX on Fb and EC, an introductory overview indicating that in Fb and EC signal transducer are expanded after AS- and AS-Tspan8-TEX-treatment, but transporter activators only in AS-Tspan8-TEX treated Fb and transcription regulators predominantly in AS-Tspan8-TEX-treated EC. Sorting according to protein classes revealed further differences, e.g., expansion of adhesion and ECM proteins is only seen in TEX-treated Fb, whereas expansion of transcription factors (Tf) is dominating in TEX-treated EC (Appendix A).

### 3.4. AS-TEX and AS-Tspan8-TEX Promote Signaling Pathway Activation in Lung Fibroblasts

AS-TEX promoted 121 mRNA and AS-Tspan8-TEX 459 mRNA upregulation in Fb coculture, only some mRNA being shared by AS- and AS-Tspan8-TEX treatment. STRING functional analysis for biological processes revealed metabolism and gene expression regulation being most frequently affected by AS- and AS-Tspan8-TEX treatment. A higher number of edges after AS-Tspan8- than AS-TEX treatment was observed for angiogenesis, migration, and translation regulation. Cell death regulating genes were more strikingly affected by AS-TEX treatment. It should be noted that these cellular processes regulating genes were, at least not strongly clustered with only translation regulating genes being exclusively recovered in a histone enriched cluster of AS-Tspan8-TEX-treated Fb (Figure 4A–C). For clarity of presentation, only high confidence connected genes are shown and unconnected nodes were omitted. However, as demonstrated (Appendix A), the overall impact of TEX treatment on activation of cell biology processes regulating genes was not affected by these restrictions. 

AS- and AS-Tspan8-TEX also account for Fb mRNA downregulation. A higher number of mRNA is affected by AS-Tspan8- than AS-TEX. Downregulated mRNA mostly are recovered in 3 dense clusters in AS-Tspan8-TEX-, but only two smaller clusters in AS-TEX-treated Fb (Appendix A). Overrepresented edges of reduced mRNA engaged in biological process regulation unraveled a significantly higher incidence of mRNA engaged in regulation of signal transduction and response to stimulus. Regulation of differentiation and angiogenesis were only affected by AS-Tspan8-TEX treatment, whereas cell death regulation was only overrepresented in AS-TEX-treated Fb (Figure 4D). Thus, mRNA downregulation confirmed the particular engagement of AS-Tspan8-TEX treatment in Fb reprogramming. The finding suggests that TEX-initiated signaling cascade activation may finally amplify or negotiate an initial trigger.

Notably, upregulated mRNA expression corresponds with protein expression. Thus, α6β4 expression became most strongly upregulated by AS-Tspan8-TEX treatment. In concern about migration, integrin α5 and β1 chains, CD44^1^, and Tspan5^1^ also were higher after AS-Tspan8- than AS-TEX treatment. CFL1^1^, initiating actin polymerization [59], was higher after AS- and AS-Tspan8-TEX treatment. Some signaling molecules, shown for PKCA^1^ and JNK^1^, were not upregulated. p38-MAPK^1^, RhoB^1^ [60], and particularly pERK1/2^1^ were more strongly expressed after AS-Tspan8- than AS-TEX treatment. Finally, reactive oxygen species (ROS)/angiogenesis regulating NOX1^1^ and NOX4^1^ [61,62] expression was slightly increased in AS-Tspan8-TEX-treated Fb. Tf NFκB p65^1^ expression remained unaltered, whereas predominantly AS-Tspan8-TEX promoted Jun^1^, FOS^1^, HIF1α^1^, and SMAD4^1^ upregulation. Foxo3^1^ as well as eLF1^1^, a component of the RNA polymerase II elongation complex regulating c-Myc^1^, Sox2^1^, and Oct4^1^ [63,64] were strongly elevated after AS-Tspan8-TEX treatment (Figure 4E). Some RTK, most strongly NTRK1^1^, NTRK2^1^, TYRO3^1^, and VEGFR2^1^ and less pronounced ALK^1^, ErbB3^1^, and MET^1^ became only upregulated by AS-Tspan8-TEX. AS-Tspan8-TEX also promoted Akt^1^ phosphorylation, IRS1^1^ and lck^1^ upregulation, which we missed at the mRNA level due to our restriction to an mRNA level of ≥1000 hits (Figure 4F). At the functional level, the most intense effect of AS-Tspan8-TEX was seen on Fb invasiveness, hardly promoted by AS-TEX (Figure 4G). Fittingly, COL4A^1^, VIM^1^, and ADAM15^1^ expression were increased in AS-Tspan8-TEX-treated Fb. However, MMP9^1^ expression was not affected (Figure 4H).

With the utmost strong impact of AS-Tspan8-TEX on Fb adhesion, migration, and invasion, we proceeded with network analyses of signaling molecules, proteases and apoptosis in TEX-treated Fb, where up- and downregulated mRNA was concomitantly evaluated. Due to Fb being infrequently affected by AS-TEX, providing the numbers of affected mRNA appeared preferable to the statistical analysis. For signaling molecule analyses, we differentiated between soluble factors (cytokines, chemokines, interleukins) that act at the cell surface and membrane receptors that could activate signaling cascades or may become internalized by ligand binding/activation. The third group covers some of the most prominent cytoplasmic signaling components that could become affected independently or via signaling receptors. The differences between AS- and AS-Tspan8-TEX treated Fb were far more striking than expected. Thus, from 25 signaling pathway engaged genes, only 8 mRNA (cytokines, GPCR, EGFR, MAPK pathway components including ERK1/2, and NFκB) were upregulated in AS-TEX-treated Fb. We should mention that reduced or upregulated (red circle) mRNA recovery was not taken into account, as the ratio was similar in AS- and AS-Tspan8-TEX treated Fb. Although the engagement of individual signaling components in several pathways provides a hindrance in selective pathway assignment, this becomes irrelevant in view of the overall strongly reduced capacity of AS-TEX in provoking signaling responses (Figure 4I–K).

There remained the question of a contribution of TEX to motility/invasion by proteolysis and/or apoptosis [65,66]. Distinct to signaling, expression of too few apoptosis engaged and proteolytic molecules was altered in AS-TEX treated Fb for sustaining overrepresentation. Nonetheless, AS-TEX treated Fb contained a significant number of apoptosis regulating molecules, the majority being upregulated. Instead, altered death receptor signaling molecules were only seen in AS-Tspan8-TEX treated Fb (Appendix A). Finally, expression of proteases is rarely altered by AS- or AS-Tspan8-TEX treatment in Fb, and less than 50% become upregulated. However, it is notable that in AS-TEX-treated Fb the majority of affected proteases is engaged in signaling and that exopeptidases are overrepresented only in AS-Tspan8-TEX-treated Fb (Appendix A). 

AS-Tspan8-TEX promoted matrigel invasion more strongly than AS-TEX. This is accompanied by stronger upregulation of signaling-related molecules, including soluble factors, membrane receptors and, less pronounced, intracellular signaling pathway components. We interpret the findings that predominantly AS-Tspan8-TEX binding promotes ligand clustering and activation.

Taken together, integrins, selectins, matrix modulating enzymes and matrix proteins are more strongly affected in AS-Tspan8- than AS-TEX-treated Fb. This might explain the impact of AS-Tspan8-TEX on embedding and outgrowth of migrating tumor cells in the lung. Histone modulations affecting gene accessibility and subsequently transcriptional activity may account for the strong impact on metabolic processes, where the higher targeting efficacy of AS-Tspan8-TEX could additionally contribute to nutrient transport. Finally, AS-Tspan8-TEX exert a far stronger impact than AS-TEX on soluble factor- and receptor-initiated signaling cascades. Whether this is accompanied by a shift towards the phenotype of cancer-associated Fb (CAF) in vivo, which initiate a feedback response of the tumor cells, remains to be explored [67]. Irrespective of this open question, the unexpectedly strong impact of AS-Tspan8-TEX on soluble factor and surface receptor signaling hints towards central importance of TEX binding-initiated signaling as a central hub in Fb remodeling.

### 3.5. AS-Tspan8-TEX Promoted Endothelial Cell Maturation and Activation

AS-Tspan8-TEX promote EC growth, spriting and tube formation (Figure 5A) [46,50,54,68,69,70]. Because distinct to Fb, expression of only few mRNA became upregulated by coculture with AS-TEX, we searched only for clusters of upregulated genes in AS-Tspan8-TEX-treated EC. Despite this restriction, overrepresented edges were only seen in UniProt Keywords covering chromosomes, nucleosome cores, the nucleus and DNA binding genes mostly recovered in one strong histone cluster. A less dense cluster contained cytokines/chemokines or receptors (Figure 5B).

As the few RNA upregulated after AS-Tspan8-TEX treatment contained cytokines, we controlled at the protein level cytokine/chemokine expression. Data are presented as fold increase compared to untreated EC. Though expression levels were mostly very low, the vast majority of chemokines, cytokines and growth factors displayed at least a 3-fold increase after AS-Tspan8-TEX treatment (Figure 5C). Controlling for additional chemokine receptors by WB showed strong upregulation of the chemokine receptor CXCR2^1^, a ligand of CXCL1, -2, -3, -5 and -6^1^ [71] by AS-Tspan8-TEX, which was confirmed by coimmunoprecipitation of CXCR2 with CXCL1. CXCR4^1^, the major ligand of CXCL12^1^ [72], and the CXCL13^1^ ligand CXCR5^1^ [73] were exclusively, though weakly, upregulated by AS-Tspan8-TEX. PI3K-p85^1^, a most common downstream signaling component of chemokine receptor activation [74], was upregulated only in AS-Tspan8-TEX-treated EC. This also accounted for the phosphoprotein PXN^1^ that acts as a scaffold in focal adhesion via binding to signaling and structural tubulin, actin and vimentin proteins [75] (Figure 5D). WB additionally confirmed VEGF and VEGR2 upregulation by AS-Tspan8-TEX. Instead, AREG^1^, belonging to the EGF^1^/TGF^1^ growth factor family [76], was high in AS-, but reduced in AS-Tspan8-TEX-treated EC. Of note is the upregulated VCL^1^ expression, which under force stress conditions is recruited towards CDH5^1^ (VE-cadherin) stabilizing EC junctions [77]. Finally, NRF2^1^, which under stress migrates to the nucleus regulating antioxidant response element transcription [78], becomes upregulated by AS-Tspan8-TEX treatment. In the resting state, it is associated with KEAP1^1^ that is also upregulated in AS-Tspan8-TEX-treated cells, the complex together with ubiquitin ligase accounting for NRF2 degradation [79] (Figure 5E).

The unexpectedly high increase of chemokines, -receptors and growth factors prompted us searching again for overrepresented nodes in AS- and AS-Tspan8-TEX-treated EC, but using less stringent selection criteria (signal intensity >500, 1.5-fold increase). Although not recovered in clusters, which is shown for AS-TEX-treated EC (Appendix A), STRING functional protein analysis uncovered overrepresentation of regulation of several biological process preferentially in AS-Tspan8-TEX-treated EC like chromosome organization, transcription and RNA metabolism, others were more strongly affected in AS-TEX-treated EC, e.g., response to stimulus and differentiation (Figure 5F). Using the same less stringent criteria for reduced mRNA in AS- or AS-Tspan8-TEX-treated EC uncovered the highest number of overrepresented nodes in AS- and AS-Tspan8-TEX-treated EC being engaged in regulation of metabolism, followed by gene expression. Overrepresented nodules for chemotaxis and coagulation were only seen in AS-TEX-treated EC. Overrepresented edges for regulation of translation were restricted to AS-Tspan8-TEX-treated EC (Appendix A, Figure 5G).

Like for TEX-treated Fb, we proceeded with in silico analysis of selected topics. However, with the comparably low number of genes that expression became altered by TEX treatment of EC, up- and downregulated mRNA were concomitantly evaluated, indicating upregulated mRNA by a red circle. Searching for signaling related molecules and pathways, revealed a higher incidence of nodes for receptor signaling in AS-Tspan8-TEX-treated EC, but a dominance of receptor signaling regulating proteins in AS-TEX-treated EC. On the other hand, cytokines/chemokines, signaling receptors and intracellular signaling molecules were only enriched in AS-Tspan8-TEX-treated EC. Furthermore, the vast majority of overrepresented edges (41/45) after AS-TEX treatment was downregulated, while 59/102 mRNA were upregulated in AS-Tspan8-TEX-treated EC (Figure 5H). Thus, one central difference between AS- versus AS-Tspan8-TEX treatment relies on AS-TEX frequently hampering signaling molecule and pathway activation.

In view of previous findings, we expected upregulated vasculo-/angiogenesis-related mRNA recovery in AS-Tspan8-TEX-treated EC, the accompanying lethal thrombosis being classified as a secondary phenomenon [50,80]. A higher number of up- and downregulated vasculo-/angiogenesis-related genes were seen after AS-Tspan8- than AS-TEX treatment, altered mRNA expression seen in both TEX-treated EC being underlined and inhibitory genes being shown as open bars. Genes interfering with EC-associated smooth muscle (SM) activity are included (Figure 5I). String functional protein analysis of overrepresented edges confirmed the striking discrepancy between the impact of AS- and AS-Tspan8-TEX treatment. Overrepresented edges were seen exclusively or at a far higher significance level after AS-Tspan8-TEX treatment and presented a wide range of activities from vasculogenesis to EC sprouting and vascular SM profileration, to name a few (Figure 5J). Stimulated by the disparity in angiogenesis-related gene modulation by AS- versus AS-Tspan8-TEX treatment, we finally search for overrepresented signaling molecules/pathways selectively in angiogenesis-related genes. Strong differences were seen for RTK signaling overrepresented edges. Small GTPase signaling was only overrepresented in AS-Tspan8-TEX treated EC (Figure 5K). The striking impact of AS-Tspan8-TEX on EC mRNA including the impaired EC-layering SM development may well explain pronounced vasculo-/angiogenesis and facilitate tumor cell immigration and emigration.

Finally, Uniproct-Keyword overrepresentation of nuclear components (Figure 5B) provoked searching for transcription engaged mRNA in AS- and AS-Tspan8-TEX treated EC. Only 3 transcription-related mRNA became upregulated by AS-TEX, but 44 by AS-Tspan8-TEX treatment, with an >3-fold increase of e.g., Bcl11a^1^, Foxe1^1^, Hes7^1^, Irf7^1^, and Zfp787^1^ (Figure 5L). This strong difference was sustained by the recovery of overrepresented edges, where beside a significantly lower number of mRNA affected by AS- than AS-Tspan8-TEX the striking difference in chromatin binding and the absence of DNA and protein containing complex binding RNA in AS-TEX-treated EC should be noted. In addition, only one mRNA (Six3) was upregulated in AS-TEX, but 27/58 in AS-Tspan8-TEX treated EC (Figure 5M).

The EC signaling cascade activation by TEX provided interesting results. First to note, chemokine and growth factor activation strikingly depended on AS-Tspan8-TEX. Second, the majority of angiogenesis-related mRNA were exclusively upregulated in AS-Tspan8-TEX-treated EC. Furthermore, though AS- and AS-Tspan8-TEX are engaged in signaling regulation, Tf and transcription-initiating molecules were rather exclusively upregulated in Tspan8-TEX-treated EC. We suggest that these mRNA in concert with the abundance of angiogenic growth factors suffice for vasculo-/angiogenesis induction by AS-Tspan8-TEX. The strong impact of AS-Tspan8-TEX in readout systems supports our hypothesis. Finally, clustering of histone genes was only seen in AS-Tspan8-TEX-treated EC. This unexpected finding stimulated us to more closely peer miRNA and miRNA regulation in AS-Tspan8-TEX-treated EC.

### 3.6. TEX miRNA Upregulation and Affected Targets in Endothelial Cells

There is ample evidence that EV miRNA can support angiogenesis [81]. Only few miRNA becoming up- or downregulated in AS- or AS-Tspan8-TEX-treated EC (Appendix A), facilitated the search for predicted targets, which were selected according to http://www.microrna.org and http://www.targetscan.org programs and were correlated with the expression in TEX-treated EC. The limit was set to a signal strength of ≥500 and an ≥1.5-fold difference to untreated EC.

Selecting for predicted targets of upregulated miRNA after coculture with AS- and/or AS-Tspan8-TEX revealed reduced expression of 24% of 282 predicted mRNA targets. The percent of downregulated mRNA varied for the individual miRNA, miR-150 and miR-181a showing the highest rate of inverse mRNA versus miRNA expression. Notably, few RNA displayed higher expression levels despite increased miRNA recovery (Figure 6A). STRING analysis of predicted downregulated mRNA in AS-TEX-treated EC displaying increased miRNA revealed some connectivity around Mapk1^1^, Mapk9^1^, Map2k1^1^, and Kras^1^ that were not seen in the net of confirmed targets. Accordingly, overrepresented edges were only recovered for regulation of cell communication, signaling and gene expression (Figure 6B). In AS-Tspan8-TEX-treated EC pronounced connectivity of predicted downregulated mRNA correlating with upregulated miRNA also focused around MAPK and included TP53^1^, Nras^1^, and Prkcd^1^. The cluster and the central nodes were not seen in the net of confirmed downregulated mRNA, overrepresented nodes being only confirmed for regulation of signaling and cell communication (Figure 6C). It should be mentioned that for the sake of clarity unconnected genes that expression was not affected are not shown in Figure 6B,C.

Predicted mRNA recovery in EC showing reduced miRNA expression after coculture with AS- or AS-Tspan8-TEX (Appendix A) argues against the reduced miRNA being accompanied by mRNA release from repression. mRNA expression remained mostly unaltered and few changes were equally distributed between up- and downregulated mRNA (Appendix A). Finally, STRING functional protein analysis revealed no enrichment for upregulated mRNA targets of reduced miRNA (Appendix A).

Though miRNA recovery in TEX did not correspond to miRNA recovery in TEX-treated EC, in >70% expression of miRNA in TEX-treated EC inversely correlated with the mRNA recovery, the 44 downregulated mRNA being listed (Figure 6D,E). This suggested functional competence of the affected miRNA. First, miRNA target binding was proven by luciferase activity of 293T cells transfected with the pMIR-luc-UTR reporter plasmid containing BTG2 3′UTR, the 3′-UTR of the tumor suppressor Btg2^1^ [82] allowing for miR-146b binding. In the presence of the BTG2 3′UTR containing luciferase reporter, luciferase activity was significantly reduced (Figure 6F). Functional activity of miRNA recovery was also controlled by coculturing EC with miR-181a and miR-146b inhibitors, both miRNA being high in AS-Tspan8-TEX-treated EC. Coculturing EC for 48 h with miR-181a or miR-146b inhibitors promoted IL1α^1^, weakly IL1β^1^, and strongly VEGF^1^ expression, which are targets of these miRNA. Release from repression was accompanied by increased EC tube formation, when miRNA inhibitor transfected EC were seeded on precoated matrigel (angiogenesis assay kit) (Figure 6G,H). Thus, TEX coculture-promoted increased miRNA displays functional activity.

Only very few predicted targets of upregulated miRNA in AS- or AS-Tspan8-TEX-treated EC were confirmed in TEX-treated EC, which according to STRING network analysis did not form clusters, and overrepresented edges were rare. Nonetheless, target cell miRNA was function competent.

Target cell-inherent regulation of miRNA expression in TEX-treated EC demanded for a search towards possible pathways. One engaged player could be long nc (lnc)RNA.

### 3.7. TEX-Induced Changes in lncRNA and the Suggested Contribution to Transcription and miRNA Regulation

Long noncoding RNA (lncRNA) and pseudogenes are functionally important elements of the genome [83,84,85]. There were 33 lncRNA and 164 nc pseudogenes recovered in rat Fb and EC (Appendix A), from which 26 and 25 were recovered at a higher, and 22 and 25 at a lower level in AS-Tspan8-TEX than untreated EC and Fb (Appendix A). Differences between lncRNA in EC and Fb were rare (Appendix A). Searching, whether AS-TEX treatment affects lncRNA expression revealed 4, respectively, 6 significant changes in EC and Fb (Appendix A, Figure 7A). Instead, 14 EC and 34 Fb lncRNA expressions were altered after AS-Tspan8-TEX treatment (Appendix A, Figure 7B). TEX coculture-induced changes rarely fitting to lncRNA expression in TEX (Appendix A versus Appendix A), argues against a significant contribution of lncRNA transfer from TEX into targets. Thus, the significantly stronger impact of AS-Tspan8- than AS-TEX on target cell lncRNA points towards an engagement of TEX Tspan8/Tspan8 complexes also on lncRNA regulation.

There remained the question on the functional activity of lncRNA that expression became altered by TEX treatment. Unfortunately, only for 14 lncRNA that expression was up- or downregulated in AS-Tspan8- or AS-TEX-treated EC and Fb hints towards possible functions are reported (Appendix A). There were 5 lncRNA described acting at the chromosome level, 4 affecting transcription, 4 interfering with miRNA and 7 contributing to protein complex formation including RNA binding proteins (RBP). Two lncRNA are engaged in all these processes (Figure 7C). Though available data are not sufficient for drawing firm conclusions, the high impact of AS- and AS-Tspan8-TEX on chromosome structure as well as the inefficacy of some miRNA to affect mRNA expression could possibly rely on lncRNA.

To sustain this hypothesis and in view of more abundant information on human lncRNA, we searched for altered lncRNA in a human Tspan8kd tumor line after wt TEX treatment. From 185 lncRNA, 60 were recovered at a comparable, 78 at a reduced, and 47 at a higher level in Tspan8kd cells than wt-TEX. Coculture with wt-TEX promoted 54 lncRNA upregulation and 22 lncRNA reduction in Tspan8kd cells (Appendix A, Figure 7D). For 49 lncRNA disease associations were described, and for 38 lncRNA (upregulated: 29, reduced: 9) some annotations are known or suggested (Appendix A). Trying to integrate this information in the mode of lncRNA activities points towards a dominance of chromosome modulation (27), followed by the most frequently cited sponging of miRNA (24). lncRNA also are engaged in protein complex formation (18), the complexes frequently contain RBP and are involved in splicing and RNA translation. However, lncRNA also associates with single proteins affecting directly their activity or transport. Finally, these lncRNA are engaged in transcription initiation (5). Notably, shuttling between the nucleus, mitochondria, and the cytoplasm was described for 23 lncRNA, where lncRNA fulfill location-dependent distinct activities [86] (Figure 7E).

According to the current state of information, lncRNA is preferentially engaged in chromosome modulation and complex formation with proteins frequently affecting splicing and translation. With all caution due to limited information, lncRNA activity is by no means restricted to competitive endogenous (ce)RNA activity [87]. Instead, they are of major importance in chromosome organization including Tf regulation [84] and, notably, TEX treatment affects recovery of lncRNA in targets at a strikingly higher rate than that of RNA or miRNA.

Taken together, four features appear of particular interest for non-transformed target modulation by TEX. First, TEX activate target cell inherent programs. Second, one of the central hits in non-transformed cells are metabolic shifts, which can affect histone modulations promoting or suppressing transcription. Third, at least partly tetraspanin-dependent, TEX promote signaling cascade activation. Fourth, besides displaying ceRNA activity, and also partly Tspan8-TEX-dependent, changed lncRNA expression and location apparently is of major importance in regulating gene accessibility and translation.

## 4. Discussion

TEX received much attention as transferring tumor-growth and -progression promoting activities into non-cancer-initiating cells (Non-CIC) [88] as well as host cells promoting (lymph)angiogenesis [7,89], premetastatic niche establishment [8,9,10], and deviation of hematopoietic cells towards immunosuppression [90]. We recently elaborated the mode whereby CIC-TEX affect Non-CIC, which revealed that an impact of TEX-Tspan8 predominantly relies on the engagement of Tspan8 complexes in TEX biogenesis [5,30,54,57,91] including the transfer of Tspan8-associated CIC-markers into TEX and, most importantly, on Tspan8 facilitating TEX binding and uptake by target cells via ligands for Tspan8-associated molecules [5,27,38,52,55,68]. Uptaken TEX stimulating non-CIC inherent programs [5,30,54] and Tspan8 being particularly engaged in TEX binding and uptake, we asked whether non-transformed cells respond alike oncogene-transformed cells. We selected for Fb that contribute to premetastatic niche generation [9,92] and EC responding to TEX after 1h to 5d [50], which we considered a hint for a target cell autonomous response. To keep the continuity with previous work, we retained AS-Tspan8 cells as TEX donor and controlled for the selective Tspan8 contribution by coculturing EC and Fb also with AS-TEX. We confirm TEX promoting target cell autonomous program activation. Accordingly, EC and Fb respond distinctly.

### 4.1. The mRNA and miRNA Profile of EC, Fb, and TEX

In advance of analyzing the impact of TEX on EC and Fb, mRNA with a signal strength of >1000 was analyzed to provide hints towards a possible contribution to molecular activities. EC and Fb displayed minor differences, which also accounted for TEX. The lower number of mRNA overrepresented in TEX compared to cells is in line with other reports. A significant enrichment of mRNA encoding E2F transcriptional targets and histone proteins points towards originating from a cytoplasmic RNA pool, packed into EV during S-phase [93].

Similar conclusions were drawn from the miRNA analysis. Few miRNA were recovered at a higher level in TEX than cells, which fits to a selective transfer of miRNA into ILV/MVB, where GW182 bodies, possibly in association with miRNA-loaded AGO2 and the RNA-induced silencing complex (RISC), are sorted into MVB for secretion and/or lysosomal degradation [19,94]. The unexpectedly low number of miRNA enriched in AS/AS-Tspan8 TEX points towards a paucity of proteins engaged in recruitment into MVB. One of the missing components in AS/AS-Tspan8 cells could be CD44v6, which contributes to miRNA recruitment [30,54]. Additional proteins engaged in miRNA transfer into ILV remain to be elaborated. Finally, in view of our question towards the contribution of Tspan8 to TEX message transfer, it also was important to compare the miRNA profile of AS- and AS-Tspan8-TEX. The reliability of the few distinctly recovered miRNA was seconded by 19 of 26 miRNA enriched in AS-Tspan8-TEX being reduced in ASML-Tspan8kd-TEX. 

Taken together, the overall comparability of EC, Fb, and TEX mRNA and miRNA as well as the few TEX-promoted changes facilitated uncovering the mode of TEX activity and the functional assignment of engaged genes.

### 4.2. Is There a Selective Contribution of Tspan8-TEX to the Target Cell Response? 

Both Fb and EC respond to TEX, exhibiting a significantly higher response to AS-Tspan8- than AS-TEX. The stronger target cell remodeling by AS-Tspan8-TEX is not or only to a minor degree due to the TEX cytoplasm that components rarely are recovered in the target cells. This includes TEX mRNA processing complexes. It is discussed that the RNA processing machinery is efficiently transferred into EV, and that EV uptake-promoted target cell reprogramming might be mediated by the EV-derived components of the RNA processing complexes [95,96]. However, this did not account for our model system. Instead, Tspan8-complexes in the TEX membrane may be decisive, as repeatedly reported for TEX tetraspanin complexes. CD81 complexes on DC, but also T cells account for uptake by hematopoietic cells, Tspan8-α6β4 for uptake by lung fibroblasts and Tspan8-α4/α5 for uptake by EC. Tspan8, CD151, CD9, and CD181 also contribute to uptake by several tumors [46,50,97,98,99]. Though not essentially required, target cell tetraspanin webs can assist TEX uptake [27], where Fb and EC express CD81, Tspan31, Tspan4, and above all CD63 at a high level. We have not excluded an additional selective contribution of these target cell tetraspanins, but it may be minor taking the strong impact of TEX-Tspan8. In line with the importance of tetraspanins/tetraspanin complexes in initiating target cell remodeling is the particularly strong contribution of target cell membrane-integrated receptors.

Thus, the major contribution of AS-Tspan8-TEX to target modulation relies on Tspan8 complexes facilitating binding and uptake, where we suggest a binding-induced hub requiring particular attention. CD9, highly expressed in AS-TEX, may fulfill a similar role, but with far lower efficacy. 

### 4.3. The Strong Response of Fb to TEX and the Dominance of Signaling Cascade Activation

Searching for the impact of AS- and AS-Tspan8-TEX on EC and Fb revealed about 4-times stronger responses of Fb than EC. This was unexpected. It might rely on high expression of ICAM1 that beside others binds Tspan8. A possible contribution of highly expressed ICAM5^1^, integrins α1^1^ and β5^1^ and several galectins remains to be explored. 

Irrespective of this open question, at the protein level, expression of lung premetastatic niche supporting α6β4 as well as of several RTK, Akt, lck, MAPK, and RhoB was linked to AS-Tspan8-TEX treatment that may concomitantly support motility with a possible linkage via RhoB facilitating the traffic of src to the cell membrane and of Akt to the nucleus [60]. Increased NOX1 and NOX4 expression could predominantly cope with ROS during premetastatic niche preparation in advance of angiogenesis induction [61,62]. Of particular interest also appeared the upregulated expression of eLF1, a component of the RNA polymerase II elongation complex that was identified as one of the Tf influencing the promoter activity of c-Myc, Sox2, and Oct4 [63,64]. For some additional upregulated proteins the mode of action being not yet defined or controversially discussed, we proceeded searching for overrepresented edges. Three upregulated mRNA clusters were recovered in AS- or AS-Tspan8-TEX-treated Fb. A cluster containing mostly transporters, cytosolic signaling molecules, GTPases and phosphatases was larger in AS-Tspan8- than AS-TEX-treated Fb. Fittingly, genes engaged in regulation of migration and transport were overrepresented in AS-Tspan8-TEX-treated Fb. Apoptosis regulating gene clusters were also seen in AS-TEX-treated Fb. The connectivity network analysis of downregulated genes also showed 3 strong clusters in AS-Tspan8-TEX-treated Fb that were not or hardly recovered after AS-TEX treatment. The largest cluster contained, besides others adhesion molecules, cytokines and chemokines and was linked to a second cluster enclosing several ECM components and integrins. AS-Tspan8-TEX treatment of Fb also supported angiogenesis-related gene induction. These genes being poorly connected or clustered, we hypothesize that HIF-1 signaling activation might provide one of the initial triggers [100], which remains to be confirmed. Overall, networks regulating signaling, signal transduction, cell communication, adhesion, and migration were more strongly affected in AS-Tspan8- than AS-TEX-treated Fb.

With the Tspan8-α6β4 complex contributing to premetastatic niche formation in the lung, we expected signaling to center around integrin ligands. Without excluding an important input of integrin ligands, a wider range of signaling initiators contributed such as soluble factors and membrane-integrated receptors including CAMs, RTK, and GPCR. These findings suggest that in concern about initiating signaling cascades, TEX Tspan8-complexes may well be of central importance, where downstream signaling cascade activation could be a sequel of (TEX) Tspan8-binding initiated primary events [101].

Searching whether TEX may support creating space for invading tumor cells by clustering proteases and promoting apoptotic signaling pathway activation did not point towards a major contribution and differences between AS- and AS-Tspan8-TEX were less pronounced. However, death receptor signaling and exopeptidases were only promoted by AS-Tspan8-TEX treatment. This selective engagement in affecting membrane-bound death receptors and proteases fits to TEX-Tspan8 complexes initiating signaling cascades.

In essence, Fb are highly susceptible to respond to TEX. The pronounced response to AS-Tspan8-TEX relies on Tspan8-promoted clustering of ligands, particularly adhesion molecules, RTK and GPCR, which also become targets of cytokines, chemokines, and interleukins. This facilitates downstream signaling cascade activation. Though we are missing an explanation for the utmost strong response of Fb to TEX, the data hint towards TEX binding/uptake promoting a conversion to CAF known to exert a positive feedback on tumor cells [102].

### 4.4. The EC Response to TEX: Signaling Cascade Activation and Nuclear Events

EC and EC progenitors strongly respond to AS-Tspan8-TEX, the Tspan8-α4/α5 complex providing an important contribution [50]. Based on mRNA DS and a miRNA array and using the same aortic ring derived EC line and AS-Tspan8-TEX as well as identical coculture conditions, we confirmed these findings and proceeded elaborating the underlying mechanisms.

EC hardly respond to AS-, but to AS-Tspan8-TEX, although much weaker than Fb. This is reflected by the poor recovery of overrepresented edges under stringent selection criteria, which uncovered a weak cluster enriched in cytokines and a strong cluster for histones.

Recovery of the weak cytokine cluster was confirmed by cytokine/-receptor blots including downstream signaling via PI3K [74]. Notably is also the AS-Tspan8-TEX-promoted upregulation of VEGF and VEGFR2 as well as of Nrf2, regulating under stress antioxidant response element transcription [78], while being prone for degradation in the resting state by associating with Keap and an ubiquitin ligase [79]. In view of the upregulated expression of chemokines/-receptors and their recovery in a small cluster of overrepresented edges, we searched for activation of signaling pathway components that were rather exclusively recovered in AS-Tspan8-TEX-treated EC. Instead, AS-TEX treatment was accompanied by downregulation of several signaling molecules. So far, we have no explanation for this phenomenon, which requires further exploration.

Evaluating angiogenesis-related mRNA confirmed a significantly higher number of genes being affected in AS-Tspan8- than AS-TEX-treated EC. Remarkably, in AS-Tspan8-TEX-treated EC overrepresented nodes covered genes engaged in EC development, proliferation, migration, sprouting and tube formation as well as vascular SM cell proliferation and migration. The striking disparity between the impact of AS- and AS-Tspan8 treatment on EC was also reflected by the overrepresentation of angiogenesis-engaged signaling molecules, the strongest discrepancy being seen for RTK and small GTPase signaling, the latter possibly in response to GPCR activation.

Only few predicted targets of upregulated miRNA being recovered at a reduced level in AS- and AS-Tspan8-TEX-treated EC argues against a major role of TEX treatment-promoted miRNA upregulation in EC remodeling. Searching, whether confirmed downregulated mRNA might be central components of mRNA networks uncovered only for AS-TEX-treated EC links of miR-322 targets Vegfa to Myc, Slc2a1, Adamts1, and Hbegf. In AS- and AS-Tspan8-TEX-treated EC, several mRNA were linked to the tumor suppressor Btg2 [82] that is targeted by let-7 and miR-146b. Btg2 covers a wide range of functions as transcriptional coactivator, modulator of various nuclear receptors, which cooperate with PRMT1 catalyzing arginine methylation on histone and nonhistone proteins and translation inhibitors [103], which might be of interest in concern about the strong histone cluster in AS-Tspan8-TEX-treated EC. No overrepresented nodes were seen for downregulated miRNA/upregulated mRNA in AS-Tspan8-TEX-treated EC. Thus, though function-competent, TEX-modulated miRNA might exert a minor effect on mRNA expression in EC.

Finally, AS-Tspan8-TEX treated EC present a cluster of chromosome, nuclear and DNA binding genes. This tempted us speculating on upregulation of Tf that was confirmed for 44 mRNA versus only 3 mRNA in AS-TEX treated EC. Strongly upregulated ARID3a mRNA promotes IFNα expression [104], BCL11a^1^ binds to selective histone subunits, particularly RBP4 [105], FOXE1 together with ETV1^1^ promotes telomerase reverse transcription promoter activation [106] and also affects via GLI2^1^ Wnt/β-catenin pathway activation [107]. HES7 is best known for its dominant role in somitogenesis with synergistic signaling including TBX6^1^ and Notch [108]. OXT1 overexpression is accompanied by CCND1^1^, PARP^1^, CASP3, and BAX^1^, which affect proliferation and apoptosis [109]. Presently available information supporting the engagement of Tf in shaping chromosomes and transcription initiation requires additional studies to unravel the pathways from the initial trigger to Tf activation and the consequences on mRNA availability. Nonetheless, the analysis of overrepresented edges confirmed most striking differences between AS- and AS-Tspan8-TEX treated EC in chromatin and (double stranded) DNA binding. Overrepresentation of protein containing complex binding genes and helicases were only seen after AS-Tspan8-TEX treatment.

Taken together, the strong impact only of AS-Tspan8-, but not AS-TEX, confirmed previous studies on upregulation of EC maturation-, activation- and EC-associated SM-related genes. Initiation of these genes was mostly accompanied by membrane receptor, but also chemokine activation. The second peculiarity of AS-Tspan8-TEX-treated EC relied on the appearance of histone clusters and an unexpected strong upregulation of Tf that we presently could not retrace to receptor signaling activation. Nonetheless, this histone cluster is remarkable as it may well present one of the fundaments for the multiple and wide-ranging responses to EV uptake [110,111,112,113]. As lncRNA could contribute to this histone cluster overrepresentation [114], we will summarize our findings offering some suggestions for discussion.

### 4.5. lncRNA Shuttling: Melding and Resolving the Distinct EV Activities?

The multifaceted roles of lncRNA and pseudogenes recently received ample attention [83,84,85,115,116]. Yet, functional activities and relevance largely remain to be clarified. lncRNA is engaged in histone modification, can sponge miRNA, but also can contain miRNA sequences, interferes with transcription, can bind RNA and protein complexes. So far, ce-lncRNA is most intensely explored in concern about cancer-related miRNA, where it is hypothesized to be of potential diagnostic and possibly therapeutic relevance [87,117]. The well-appreciated engagement of lncRNA in histone modification and transcriptional regulation is more complex and requires further elaboration, which also accounts for the interaction with protein complexes [84,118,119]. Pseudogenes can, but must not be protein coding. Not much is known besides their evolutionary background and some linkage to selective diseases [85,120,121]. Based on this paucity of knowledge, which is even more gravid in rats compared to humans, we hope that summarizing our observations may hint towards further progress.

First to note, changes in the availability of lncRNA in TEX-treated Fb and EC reflects target cell inherent activities rather than the TEX content. In the rat model, coculture-initiated changes in lncRNA mostly contributed to protein complex formation, followed by lncRNA affecting chromosome organization or acting as (co)transcription factors or ceRNA. Two lncRNAs contributed to all four activities. An analysis of lncRNA in TEX-treated human tumor cells provided similar results with a dominance of chromosome modulation, followed by miRNA sponging, engagement in protein complex formation frequently including RBP and being involved in splicing and translation, and least frequently in transcription initiation. Finally, the vast majority of these lncRNAs are engaged in several functions, which is well in line with lncRNA shuttling between the nucleus, mitochondria, and cytoplasm [82,122,123,124,125,126].

The unexpectedly high impact of TEX on histones as well as the upregulated expression of some mRNA targets in Fb and EC despite a TEX-promoted increase in miRNA may well be due to lncRNA activities. TEX-promoted changes in lncRNA have a striking impact on chromosome accessibility. Importantly, lncRNA shuttling between the nucleus, mitochondria, and cytoplasm could easily pave the path for activation of signaling cascades and be preparatory for transcription factor activation and translation/splicing complex assembly. Last, but not least, TEX affect target cell lncRNA at a far higher frequency than mRNA or miRNA, where we, so far, have no explanation for the higher impact of AS-Tspan8- than AS-TEX in the described model.

We have not explored the location of the lncRNA that expression differs after TEX treatment. Taking the rather equal contribution of lncRNA described for histone modeling, blocking, or promoting transcription initiation, for RNA-protein complex formation and for miRNA sponging, it appears likely that nuclear, mitochondrial, and cytoplasmic lncRNAs are equally involved. Taking the stronger impact of AS-Tspan8-TEX, which preferentially initiates target cell changes by membrane receptor activity, we hypothesize that changes in lncRNA recovery are a secondary phenomenon. However, according to the demand-oriented shuttling of lncRNA, TEX-initiated target cell reprogramming may be maintained for prolonged periods, which, at least in vivo, could far exceed the availability of TEX. If these hypotheses hold true, the “secondary” lncRNA response to TEX treatment could take over the lead role in target modulation by TEX.

## 5. Conclusions and Outlook

TEX strongly affect host Fb and EC, with Fb being more susceptible than EC. There is strong evidence for a selective contribution of AS-Tspan8-TEX on non-transformed Fb and EC target cell reprogramming. This lead role of Tspan8 relies on tetraspanin/Tspan8 complexes on TEX allowing for binding of multiple target ligands, ligand clustering strengthening binding and facilitating uptake. In line with this, both Fb and EC, respond more vigorously to AS-Tspan8- than AS-TEX. Importantly, TEX promote cell autonomous program activation, where due to the target cell’s armament, particularly of membrane-integrated proteins, Fb and EC display distinct responses. AS-Tspan8-TEX targeting a wide range of predominantly membrane-integrated receptors, but also receptor-bound soluble factors and matrix proteins, the initial triggers progress towards downstream signaling molecule activation, which can affect the target cell’s metabolism, adhesiveness/migration, apoptosis susceptibility, and frequently, culminates in downstream Tf activation. Finally, the impact of TEX on coding genes is boosted by strong bearing on lncRNA that affects histone modifications, splicing, translation, endogenous RNA competition, and protein complex organization. Due to the demand-oriented shuttling, lncRNA could lastingly modulate chromosome accessibility, transcription, translation and miRNA activity (Figure 8).

Target cell remodeling by TEX relying on host cell autonomous program activation strongly changes the view on the proposed therapeutic use of TEX, necessarily requiring an in depth analysis of distinct target cell responses including a comprehensive exploration of DNA and RNA regulation by lncRNA.

## Figures and Tables

**Figure 1 cells-09-00319-f001:**
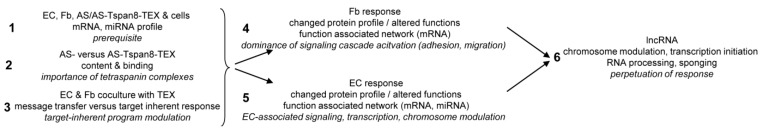
Experimental workflow.

**Figure 2 cells-09-00319-f002:**
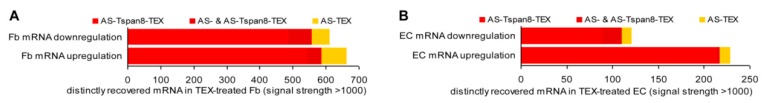
Distinctly recovered mRNA in fibroblasts and endothelial cells after coculture with tumor cell-derived small extracellular vesicles (TEX). Endothelial cells (EC) and fibroblasts (Fb) were cocultured for 2 d–3 d with 30 µg/mL BSp73AS (AS)- or AS-Tspan8-TEX. mRNA were isolated and subjected to deep sequencing (DS). (**A**) Fb mRNA and (**B**) EC mRNA ≥2-fold up- or downregulated by coculture with AS-, AS-Tspan8- or both AS- and AS-Tspan8-TEX. Fb mRNA responds significantly more frequently to coculture with TEX than EC mRNA. Both respond more abundantly to AS-Tspan8- than AS-TEX. AS- and AS-Tspan8-TEX not differing significantly points towards TEX stimulating target cell autonomous programs.

**Figure 3 cells-09-00319-f003:**
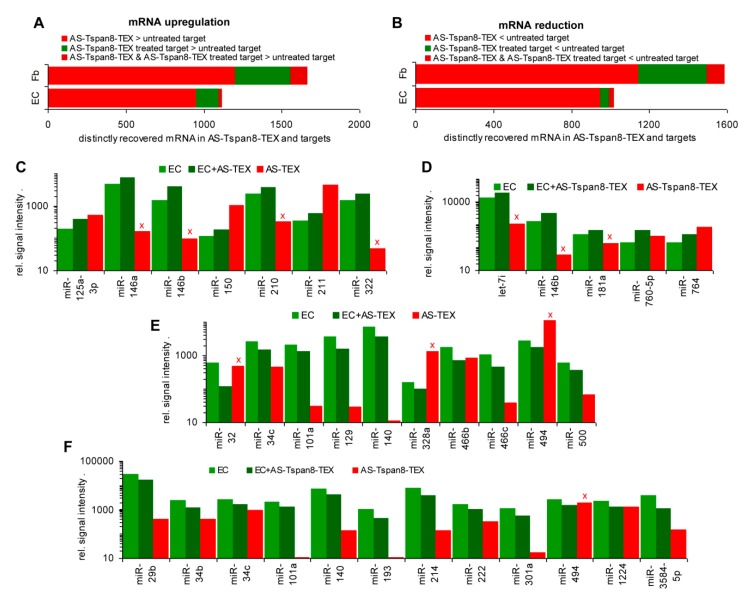
mRNA and miRNA in TEX, TEX-treated fibroblasts, and endothelial cells in comparison to untreated cells. (**A**) higher mRNA recovery in AS-Tspan8-TEX, AS-Tspan8-TEX-treated Fb or EC, or both TEX and TEX-treated Fb or EC than in untreated target cells (mRNA signal strength of ≥1000, ≥2-fold differences). (**B**) Corresponding analyses to (**A**) for reduced mRNA recovery. (**C**,**D**) miRNA upregulated in AS- or AS-Tspan8-TEX-treated EC; (**E**,**F**) miRNA downregulated in AS- or AS-Tspan8-TEX-treated EC (signal strength ≥500, ≥2-fold change), (**C**–**F**) Expression in TEX is included for comparison. miRNA expression opposing the recovery in TEX-treated EC is indicated by a red x.

**Figure 4 cells-09-00319-f004:**
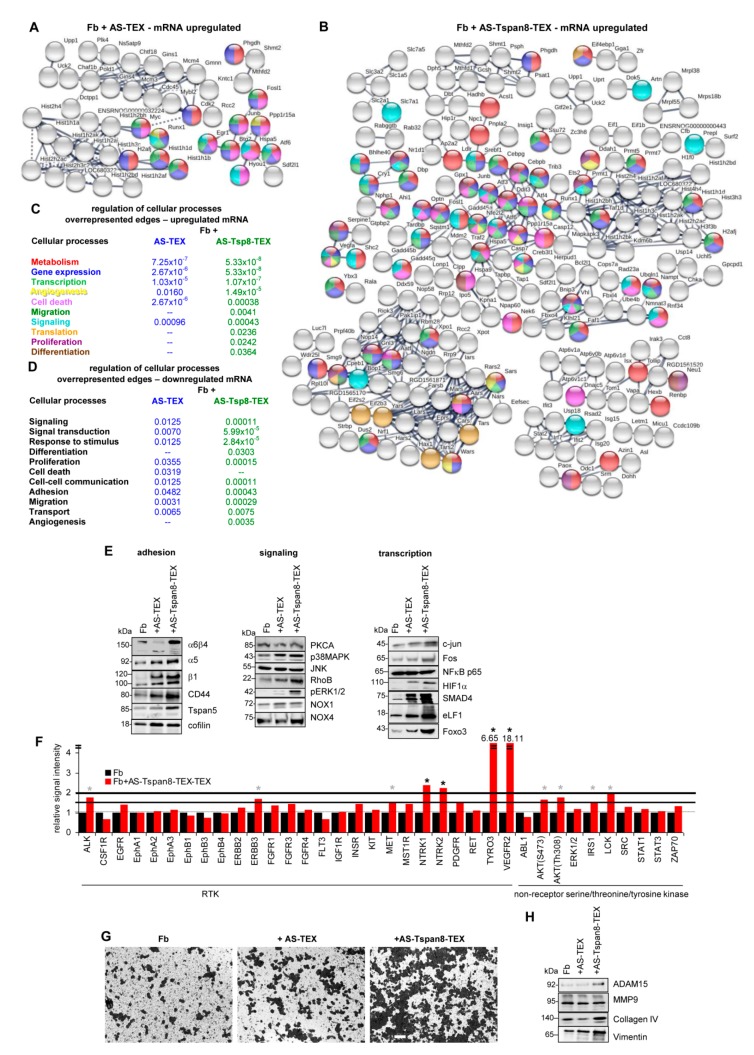
Signaling pathway integration of mRNA up- or downregulated in TEX-treated fibroblasts. mRNA was evaluated in AS- and AS-Tspan8-TEX-treated Fb for up- or downregulation compared to untreated Fb. (**A**–**D**) STRING functional protein analysis included mRNA with a signal strength ≥1000 and ≥2-fold difference between untreated and TEX-treated Fb, only connected nodes being shown (**A**) mRNA upregulated in AS-TEX- and (**B**) AS-Tspan8-TEX-treated Fb; (**C**) engagement of overrepresented edges of upregulated mRNA in the regulation of biological processes; (**D**) engagement of overrepresented edges of downregulated mRNA (see Appendix A) in the regulation of biological processes. (**E**) WB of indicated proteins in untreated, AS- and AS-Tspan8-TEX-treated Fb; (**F**) RTK and non-RTK protein array of Fb and AS-Tspan8-TEX-treated Fb. The mean of duplicates is shown, signal intensity being adjusted to 1 for untreated Fb. A significant increase in signal strength ≥1.5-fold is indicated by a grey * and by ≥2-fold by a black *; (**G**) Fb were seeded on top of matrigel-coated transwell plates that contained RPMI/10% FCS or RPMI/30µg/mL AS- or AS-Tspan8-TEX. After 48 h, cells at the lower membrane site were fixed and stained with crystal violet, light microscopy appearance (scale bar: 20 µm). (**H**) WB of the indicated proteins in Fb, AS- and AS-Tspan8-TEX-treated Fb; (**I**–**K**) STRING functional protein analysis for signaling engaged molecules up- or downregulated in (**I**) AS- and (**J**) AS-Tspan8-TEX-treated Fb, upregulated mRNA being indicated by a red circle; (**K**) overrepresented edges for signaling-related molecules and processes, the number of mRNA in overrepresented edges is shown. Full name of gene symbols: Appendix A.

**Figure 5 cells-09-00319-f005:**
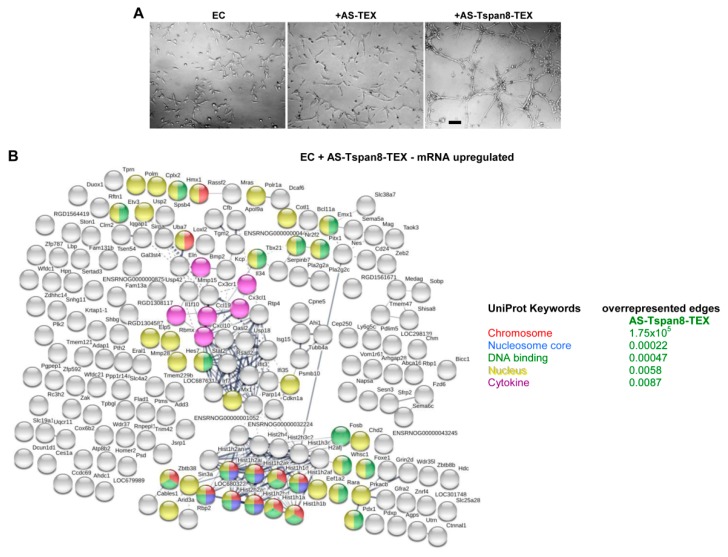
Angiogenesis-related mRNA modulation and associated signaling activation in TEX-treated endothelial cells. (**A**) EC were cultured for 48 h in matrigel in the presence of AS- or AS-Tspan8-TEX (30 µg/mL). Growth and spriting was evaluated by light microscopy (scale bar: 20 µm); (**B**) clusters of mRNA (STRING functional protein analysis) of AS-Tspan8-TEX-treated EC and indication of overrepresented edges according to UniProt Keywords; (**C**) protein array of chemokines, cytokines, growth factors, receptors, adhesion molecules, and proteases of AS- and AS-Tspan8-TEX-treated EC. Signal strength of duplicates was evaluated by imageJ and is presented adjusting signal strength of EC = 1. The signal strength particularly of cytokines being very low, only >3-fold changes between untreated vs. AS- or AS-Tspan8-TEX-treated EC were considered as significant: *. (**D**) Western blot (WB) of chemokine receptors, PI3K-p85 and paxillin in untreated and AS- or AS-Tspan8-TEX-treated EC and immunoprecipitation of CXCR2 with CXCL1 in untreated and AS-Tspan8-TEX-treated EC; lysate control and immunoprecipitation with control IgG are included; (**E**) WB of proteins playing a central role in cytokine/growth factor activation and transcription of related genes. (**F**,**G**) Overrepresented edges according to biological processes in AS- and AS-Tspan8-TEX-treated EC, STRING functional protein analysis being performed under less stringent conditions (signal strength ≥500, fold increase: ≥1.5-fold); (**F**) mRNA upregulated and (**G**) mRNA downregulated after AS- or AS-Tspan8-TEX treatment. (**H**) Signaling engaged molecules were selected by Panther Gene list. Altered signaling molecule and signaling regulating mRNA in AS- and AS-Tspan8-TEX treated EC including the incidence of overrepresented edges was evaluated by String functional protein analysis. Upregulated mRNA is indicated by a red circle. (**I**) Recovery of vasculo-/angiogenesis-related mRNA in AS- and AS-Tspan8-TEX-treated EC, genes affected by AS- and AS-Tspan8-TEX: underlined; inhibitory genes: white bars, mRNA interfering with smooth muscle development: SM. (**J**) Altered vasculo-/angiogenesis related mRNA were analyzed by STRING for engagement into selective vasculo-/angiogenesis-related functions; upregulated mRNA is marked by a red circle. (**K**) Altered signaling-related RNA recovery selectively for vasculo-/angiogenesis related mRNA was evaluated according to (**J**). (**L**) Transcription factor and transcription initiating molecule mRNA recovery in AS- and AS-Tspan8-TEX-treated EC; >2-fold differences: filled bars, >3-fold differences: red *. (**M**) Transcription-related molecule overrepresentation as evaluated by STRING; upregulated mRNA is marked by a red circle. Full name of gene symbols: Appendix A.

**Figure 6 cells-09-00319-f006:**
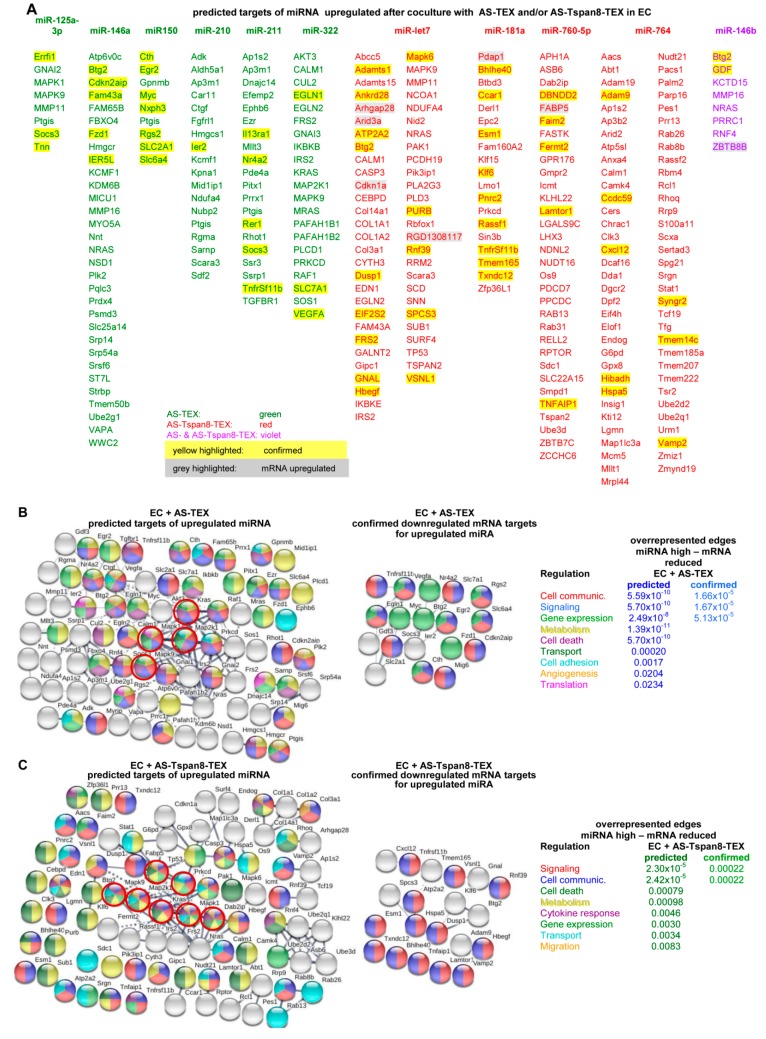
Connectivity of target mRNA of upregulated miRNA in TEX-treated endothelial cells. Predicted targets of miRNA upregulated after AS- or AS-Tspan8-TEX treatment were searched for by http://www.microrna.org and http://www.targetscan.org. (**A**) Predicted mRNA of miRNA expressed at a higher level in EC after coculture with TEX. The threshold level of mRNA was set to a signal strength of ≥500 and a ≥1.5-fold decrease; miRNA upregulated after AS-TEX treatment: green, after AS-Tspan8-TEX treatment: red, after AS- and AS-Tspan8-TEX treatment: violet; predicted mRNA reduction: highlighted yellow, predicted mRNA upregulation: highlighted grey. (**B**) Spring functional networks of predicted and confirmed downregulated mRNA corresponding to upregulated miRNA after AS-TEX treatment and some significantly overrepresented edges; (**C**) corresponding analysis to (B) for AS-Tspan8-TEX-promoted upregulated miRNA; (**B**,**C**) color of filled circles corresponds to the process of overrepresented edges; circles indicate strongly connected genes; unconnected nodes that were not engaged in the indicated processes were omitted. (**D**) Summary of mRNA recovery in AS-Tspan8-TEX-treated EC displaying increased miRNA levels and (**E**) list of downregulated mRNA; (**F**) Confirmation of BTG2 downregulation by miR-146b as revealed by a double luciferase assay. (**G**) Confirmation of miR-181a and miR-146b functional activity by culturing EC for 48 h with miR-181a or miR-146b inhibitors or a negative control inhibitor. RNA was isolated and evaluated by qRT-PCR for IL1α, IL1β, IL6, TNFα, VEGF, and TIMP1 expression; relative quantification (RQ) values±SD of triplicates, RQ value of EC cocultured with the negative control inhibitor being set as 1. A 2-fold increase was accepted as significant. (**H**) EC cultured with miRNA inhibitors as in (**G**) were seeded on matrigel-precoated plates and incubated with conditioned medium at 37 °C, 5% CO_2_ in air according to the supplier’s suggestion. Growth and tube formation was evaluated after 8h of culture by light microscopy (scale bar: 200 µm). Full name of gene symbols: Appendix A.

**Figure 7 cells-09-00319-f007:**
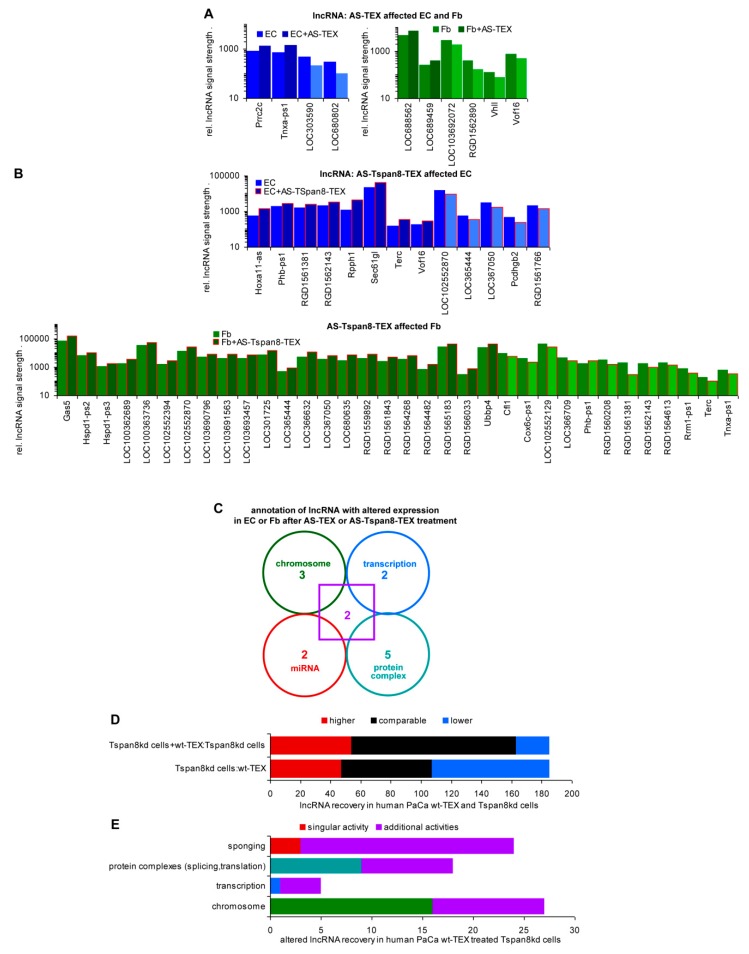
Impact of TEX-promoted long noncoding RNA (lncRNA) up- or downregulation on fibroblasts and endothelial cells. Different lncRNA recovery in EC and Fb after coculture with (**A**) AS-TEX or (**B**) AS-Tspan8-TEX. (**C**) Functional assignment of 14/52 lncRNA that expression was altered in EC or Fb after cocultures with TEX. (**D**) Comparison of lncRNA recovery in human pancreatic adenocarcinoma (PaCa) Tspan8kd cells to wild type (wt)-TEX and of wt-TEX-treated Tspan8kd cells to Tspan8kd cells. (E) Functional assignment of human lncRNA displaying altered expression in wt-TEX-treated Tspan8kd cells. Full name of gene symbols: Appendix A.

**Figure 8 cells-09-00319-f008:**
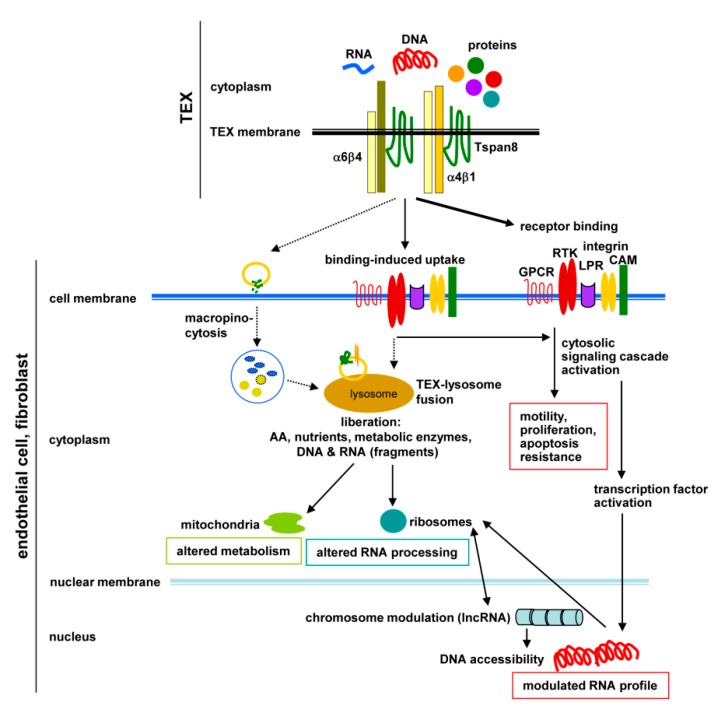
Coordinated view of Tspan8-TEX-initiated non-transformed cell remodeling. Tspan8-TEX maintain the tetraspanin-associated protein clusters, which facilitate receptor ligand binding that promotes receptor and downstream signaling cascade activation. Alternatively, receptor clustering and activation induces internalization with rarely observed for tetraspanin, subsequent exosome digestion in lysosomes and reutilization of the released breakdown product. Persistence of Tspan8-TEX promoted target cell modulation relies predominantly on activated transcription factor activity modulating the RNA profile. TEX activity is strongly supported by lncRNA that affects chromosomes/DNA accessibility and by forward-backward shuttling between nucleus, ribosomes, and cytoplasm creates circles maintaining remodeling.

## Data Availability

miRNA arrays are deposited at GEO (http://www.ncbi.nlm.nih.gov/geo/query/acc=GSE120185). Deep sequencing mRNA are deposited at ENA database (accession No: PRJEB25446).

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
