# Peer review of "Tspan8-Tumor Extracellular Vesicle-Induced Endothelial Cell and Fibroblast Remodeling Relies on the Target Cell-Selective Response"

_cells, 2020, doi:10.3390/cells9020319_

Round 1
Reviewer 1 Report
The authors have attempted to characterise the exosome populations produced by 'wild-type' tumour cell lines, a Tspan8-overexpressing cell line and a knockdown cell line, and the effects of these exosomes on two cell types that may form the premetastatic niche. Very little is made of the data obtained from the knockdown cell line, which is buried in the supplementary tables/figures and barely commented upon.
The ms is very complicated, and attempts to make too many conclusions. The authors could make two very interesting papers from this ms, one on the role of Tspan8 on exosome formation/loading, and a second on the effects of these exosomes on the target cells. This would make full use of the datasets, allow more meaningful conclusions to be drawn, and be much more accessible for readers.
My primary recommendation is for the authors to re-submit two related ms. A secondary recommendation would be to improve the quality of the English used, which sometimes obscures comprehension e.g. points iv and v in the Abstract.
Author Response
We cordially thank the reviewer for the careful evaluation of the manuscript, the helpful suggestions, and the overall positive judgment. We followed your suggestions as outlined below. All changes are indicated in red.
We apologize for not taking into account the suggestion of splitting the manuscript into two parts. The reason for our decision relies on having dealt with the suggested part one on the role of Tspan8 on exosome formation and loading since many years, those data being already published. Thus, we think we may drop into repetition at several points, which we wanted to avoid. Previous manuscripts dealing with this topic are cited (ref.27,38,46,48,50,52). Furthermore, in concern about the spare presentation of knockdown cells, this topic has been explored in more detail in ref.47 and 52 for rat cells and in ref.54 and 91 for human cells (line 249-254). In the latter manuscript it is outlined in detail that by maintenance of the tetraspanin web during internalization and vesicle biogenesis changes in the profile of small extracellular vesicles in dependence on tetraspanin expression is a secondary phenomenon due changes provoked by associated molecules. This accounts, for example, for the enrichment of some miRNA, which are actively recruited into ILV by CD44v6, whereas Tspan8 is not involved in ILV loading (mentioned in the Discussion, line 687-688, ref.30,54). For the same reason, we preferred to present the data on Tspan8 transfected cells, which avoids changes provoked by the multiple Tspan8 associated proteins in the knockdown models. We hope that you can agree with our decision.
However, in reply to your request for improving the design, we added in the result section a flow chart, as suggested by reviewer 2 to facilitate readability. We also changed in some sections the follow up of the experiments and tried to facilitate readability of some STRING analysis by omitting non-involved, non-connected items, which is mentioned in the Figure legends. Finally, some of the STRING analysis items, particularly those that indicated no significant changes were shifted into the supplement. Thus, original Fig.3A,3B was simplified by showing only highly connected nodes in Fig.4A,4B (the original version is shown in Fig.S5A,S5B for informing the interested reader that the information was not changed by the selection criteria). The following STRING row data were shifted into the supplement: Fig.3H,3I (Fig.S5D), Fig.3O,3P (Fig.S5E,S5F), Fig4F (Fig.S6A), Fig4H,4I (Fig.S6B). Panther pathway analyses were mostly omitted (original Fig.1C-1F, Fig.2B,2C,2E,2F). Fig.6A,6B (recovery of lncRNA in cells and TEX also was shifted into the supplement (Fig.S8A,S8B).
Conclusions were rewritten (line 863-890) and an overview diagram was included (Fig.8). We would be glad if you can agree with the revised conclusions.
We also took care to improve the English language and hope that you consider the changes as appropriate.
Finally, we changed the title. Reviewer 2 pointed out that the Society of Extracellular vesicle prefers the term small extracellular vesicle (EV) rather than exosomes, as exosomes are difficult to be unequivocally differentiated from other small EV. We kept this change throughout the manuscript.
Last, we most cordially want to thank you for spending your time with reviewing the manuscript and the highly appreciated help by your suggestions. Although for the reasons outlined above, we preferred not to split the manuscript, we hope very much that you consider the revised version appropriate for publication in Cells.

Reviewer 2 Report
In the present manuscript entitled “Tspan8-tumor exosome-induced endothelial cell and fibroblast remodeling relies on target cell-selective signaling activation” the author developed an extremely detailed analysis on the effect of extracellular vesicles in the promotion of premetastatic niche. The study is well conceived with an accurate methodology and presents interesting results in the field of extracellular vesicles. I would like to congratulate the authors for a well-done job. The presented results seems to fit the requirements for publication, however, some minor comments should be addressed prior to publication.
Minor comments:
I will suggest to re-write the abstract in a more “fluent” way. In abstract (line 13) TEX refers to “tumor exosomes”, while, in introduction (line 36) appears the same acronym TEX referred to tumor cell-derived small extracellular vesicles. I will suggest maintaining this last nomenclature. I will suggest to avoid using the Exo abbreviation in the main text. 2018 ISEV guidelines suggest to use the term “small extracellular vesicles” instead of exosomes when the exact biological origin of the extracellular vesicles is not stablished. I’ll encourage the authors to follow this guideline and accept this nomenclature.( https://www.ncbi.nlm.nih.gov/pubmed/30637094 ; PMID: 30637094)
In methods section (line 111), authors should add the information about the number of cells and volume of cell culture media used for extracellular vesicles isolation. From my point of view, lines 190-197 will fit better in introduction section than in results. I will suggest to add a graphical explanation of the experimental workflow followed in the presented manuscript, as it will improve the understanding of the methodology. Seems to be some missing words in lines 576/577. From my point of view, the body text, and the figures, are too large and long. I will suggest to simplify or summarize some parts of the text as well as figures. In example, some of the string figures contains too many nodes and information to be appreciated. From my point of view simplifying some of these figures will improve the comprehension of readers.Author Response
Response to Reviewer 2
We highly appreciate the reviewer’s positive judgment of the manuscript and the helpful suggestions for further improvement. As outlined below, we have considered all comments. Changes are indicated in red to facilitate retrieval.
The abstract was rewritten. We hope, it reads more fluently. We thank in particular for the hint towards exchanging the term exosomes by small extracellular vesicles. We followed this suggestion throughout including the title of the manuscript. Solely in the Introduction, where features of exosomes as a subpopulation of small extracellular vesicles are described, we maintained the term exosomes. In this section, we also included the reference that you suggested (ref.12).
As you suggested, we added information about the number of cells and the volume of medium in Material and Methods in advance of describing the TEX isolation procedure (lines 111,112).
We thank for the hint of shifting part of the introductory sentence in Results towards the Introduction. The preferential impact of Tspan8-a4/a5 on endothelial cells and of Tspan8-a6b4 on fibroblasts is now mentioned in the Introduction (lines 85-87, ref.46-50) and was omitted in Results.
We are particularly grateful for your suggestion to start the Result section with a graphic explanation of the experimental workflow (new Fig.1). We hope that this facilitates readability and that you can agree with our suggestion.
Missing words in lines 576/577. The sentences were rewritten. I hope no words are missing in the new version (lines 540/541).
You suggested shortening and simplifying text and figures to improve comprehensibility. As we wanted to provide the reader some explanation for the follow-up of the presented data, we are afraid that despite omissions (not indicated) the length of the text was not strongly shortened. However, we omitted several items (Panther pathway analyses: original Fig.1C-1F, Fig.2B,2C,2E,2F), and shifted others into the supplement. Overloaded STRING functional association networks were either simplified by including only high stringency connected nodes and by omitting non-connected nodes, which is mentioned in the figure legend. Only for Fig.4A,4B (the original version is shown in Fig.S5A,S5B for pointing out that the central information was not changed by the selection criteria). In addition, the raw data of several STRING analyses were shifted into the supplement: Fig.3H,3I (Fig.S5D), Fig.3O,3P (Fig.S5E,S5F), Fig4F (Fig.S6A), Fig4H,4I (Fig.S6B). Furthermore, Fig.6A,6B (recovery of lncRNA in cells and TEX) was shifted into the supplement (Fig.S8A,S8B). We hope that you share our opinion that readability was improved by these changes and that you can agree with the new presentation of Results.
Finally, we hope that typing mistakes were omitted and the English language was improved to be acceptable for publication.
We thank you very much for the time you spent with reading and evaluating the manuscript and for your most helpful hints towards improving readability and intelligibility. We hope that the revised version measures up to your expectations.

Reviewer 3 Report
The authors provide interesting and very well detailed evidences on the role of Tumor-exosomes (TEX) expressing tetraspanin Tspan8 on rat fibroblast and rat endothelial cells and how they could drive cell autonomous program activation. The paper is very dense of experimental evidences and figures that, from one side reflect the scientific plan used, but from another side, make the paper hard to follow. Too many description make the paper hard tofollow and undestand. The authors should arrange and reduce it in order to make the reader more confident with the content.
Minor point: in Materials and Methods, page 3 paragraph “mRNA deep sequencing (DS) and miRNA microarray”, the autors should add the NGS methods and platform used.
Author Response
Response to Reviewer 3
We most thankfully appreciate the reviewer’s comments, which were taken into account. Changes are indicated in red and are outlined in detail below.
The reviewer is most concerned about the very dense presentation of Results including the text and figures. We tried to cope with this concern by starting Results with a graphic explanation of the experimental workflow, suggested by Reviewer 2 (new Fig.1). We hope that this facilitates comprehensibility and readability. In addition, Results were partly rearranged and rewritten to increase the traceability, how the preceding findings demanded for the following experiments. We also omitted some items (Panther pathway analyses: original Fig.1C-1F, Fig.2B,2C,2E,2F) or shifted them into the supplement. This accounts particularly for presentations that unraveled a topic to be not or not of major relevance in target cell remodeling and for raw data presentation of STRING functional association networks: Fig.3H,3I (Fig.S5D), Fig.3O,3P (Fig.S5E,S5F), Fig4F (Fig.S6A), Fig4H,4I (Fig.S6B). Other raw data STRING analyses were simplified by including only high stringency connected nodes and by omitting non-connected nodes, which is mentioned in the legends. Only for the first presentation of STRING analysis (Fig.4A,4B), we decided to maintain the original files in the supplement (Fig.S5A,S5B) as a hint that the central information(s) remained unaltered in the compressed presentation. However, should you prefer, these items also can be omitted in the supplement. We hope that you can agree with these changes.
In MM, the methods of deep sequencing and the miRNA platforms were included (lines 123-125). We hope that this information is sufficient.
We also took care of avoiding typing mistakes and of further improving English language and grammar.
Reviewing manuscripts is very time consuming, particularly when long and condensed. We very much thank the reviewer for accepting this task as well as for the helpful suggestions. We hope that the Reviewer can accept our revised version of the manuscript.

Round 2
Reviewer 1 Report
N/A